# Distribution and Chemistry of Phoenixin-14, a Newly Discovered Sensory Transmission Molecule in Porcine Afferent Neurons

**DOI:** 10.3390/ijms242316647

**Published:** 2023-11-23

**Authors:** Urszula Mazur, Ewa Lepiarczyk, Paweł Janikiewicz, Elżbieta Łopieńska-Biernat, Mariusz Krzysztof Majewski, Agnieszka Bossowska

**Affiliations:** 1Department of Human Physiology and Pathophysiology, University of Warmia and Mazury in Olsztyn, Warszawska 30, 10-082 Olsztyn, Poland; ewa.lepiarczyk@uwm.edu.pl (E.L.); pawel.janikiewicz@uwm.edu.pl (P.J.); mariusz.majewski@uwm.edu.pl (M.K.M.); agnieszka.bossowska@uwm.edu.pl (A.B.); 2Department of Biochemistry, Faculty of Biology and Biotechnology, University of Warmia and Mazury in Olsztyn, Oczapowskiego 1A, 10-719 Olsztyn, Poland; ela.lopienska@uwm.edu.pl

**Keywords:** phoenixin-14 (PNX), pig, dorsal root ganglia (DRG), neuropeptides, immunofluorescence, mass spectrometry

## Abstract

Phoenixin-14 (PNX), initially discovered in the rat hypothalamus, was also detected in dorsal root ganglion (DRG) cells, where its involvement in the regulation of pain and/or itch sensation was suggested. However, there is a lack of data not only on its distribution in DRGs along individual segments of the spinal cord, but also on the pattern(s) of its co-occurrence with other sensory neurotransmitters. To fill the above-mentioned gap and expand our knowledge about the occurrence of PNX in mammalian species other than rodents, this study examined (i) the pattern(s) of PNX occurrence in DRG neurons of subsequent neuromeres along the porcine spinal cord, (ii) their intraganglionic distribution and (iii) the pattern(s) of PNX co-occurrence with other biologically active agents. PNX was found in approximately 20% of all nerve cells of each DRG examined; the largest subpopulation of PNX-positive (PNX^+^) cells were small-diameter neurons, accounting for 74% of all PNX-positive neurons found. PNX^+^ neurons also co-contained calcitonin gene-related peptide (CGRP; 96.1%), substance P (SP; 88.5%), nitric oxide synthase (nNOS; 52.1%), galanin (GAL; 20.7%), calretinin (CRT; 10%), pituitary adenylate cyclase-activating polypeptide (PACAP; 7.4%), cocaine and amphetamine related transcript (CART; 5.1%) or somatostatin (SOM; 4.7%). Although the exact function of PNX in DRGs is not yet known, the high degree of co-localization of this peptide with the main nociceptive transmitters SP and CGRP may suggests its function in modulation of pain transmission.

## 1. Introduction

Phoenixin-14 (PNX), first described by Yosten et al. in 2013 [1], was initially linked to reproductive system control, due to the high expression of this peptide in the rodent hypothalamus and pituitary gland, where, by acting through the orphan receptor G protein-coupled receptor 173 (GPR173), PNX was found to activate gonadotropin-releasing hormone-secreting cells, leading to the release of luteinizing hormone or follicle-stimulating hormone by pituitary gonadotrophs [1,2]. Nine years after PNX’s identification, further studies revealed that the presence of this peptide is not limited to the hypothalamo-pituitary complex and an engagement of PNX in the central control of several other biological processes, including food intake, energy homeostasis, water balance and modulation of inflammation or memory and anxiety, has also been reviewed [3,4].

Regarding the presence and putative function(s) of PNX in the peripheral nervous system (PNS), it should be emphasized that currently there is a huge gap in our knowledge regarding this aspect of PNX’s role, especially in the afferent part of the PNS: the existing data, coming almost exclusively from rat models, indicate the presence of PNX-immunoreactivity (PNX-IR) structures in the spinal trigeminal tract and nucleus of the solitary tract, the superficial layers of the dorsal horn (DH), the trigeminal and nodose ganglion cells and last, but not least, in neurons located in the dorsal root ganglia (DRGs) [5]. Quite recently, PNX-IR has been detected in nerve fibers (NFs) forming a dense plexus in laminae I and II of the porcine DH [6]. Although the results of the above-mentioned study support the hypothesis that there is probably a relatively large population of PNX-IR neurons in the DRGs that project axons to the superficial laminae of the DH, there are no clear data confirming either the presence of PNX in the DRGs of individual spinal cord neuromeres or their intraganglionic distribution pattern(s). Moreover, with the exception of one mention of colocalization of PNX and substance P (SP) in rat dorsal root ganglia (DRG) cells [5], there are no data available regarding the colocalization pattern(s) of PNX with other biologically active substances in specific subpopulations of DRG neurons (small-, medium- and large-diameter cells), which are believed to be involved in the regulation of different neural activities.

Therefore, in order to create a basis for further, more detailed research on the role of PNX in the sensory part of the peripheral nervous system, the present study investigated (i) the pattern of PNX-IR sensory neuron occurrence in the DRGs of selected spinal cord neuromeres; (ii) pattern(s) of intraganglionic distribution of PNX-positive (PNX^+^) cells in the examined sensory ganglia and (iii) pattern(s) of colocalization of PNX with other markers of sensory neurons (cocaine- and amphetamine-regulated transcript, CART; calretinin, CRT; calcitonin gene-related peptide, CGRP; galanin, GAL; neuronal nitric oxide synthase, nNOS; pituitary adenylate cyclase-activating polypeptide, PACAP; SP; somatostatin, SOM) in DRG neurons at different levels of the porcine spinal cord (cervical—C; thoracic—Th; lumbar—L and sacral—S). The domestic pig was chosen, as this species is increasingly employed as an animal model in studies because of its anatomical and physiological similarities to humans concerning both the structure and function of various tissues and organs [7,8,9,10].

## 2. Results

### 2.1. Mass Spectrometry Detection of PNX in Pigs DRG

The presence of “true” PNX in porcine DRGs was analyzed in the ganglia of the L5 neuromere. The mass spectrometry analyses confirmed the identity of the peptide with a molecular weight of 1583 +/− 1.5 Da. The above study identified the fractional spectra of most of the amino acids that make up PNX: DVQPPGLKVQSDPF. The list of mass spectra of each amino acid that makes up PNX is shown in Appendix A. Below is an example of a DRG L5 chromatogram (Figure 1).

### 2.2. Distribution Pattern and Morphometrical Characteristics of PNX-Containing Sensory Neurons in Studied DRGs

#### 2.2.1. Distribution Pattern of PNX^+^ Neurons

PNX^+^ nerve cells were observed bilaterally in all the studied DRGs. The total number of PNX^+^ sensory neurons counted in all the studied DRGs per animal ranged from 16714 to 19675 (18481 ± 1562; mean ± standard deviation—SD), representing 19.7 ± 1.0% of all the sensory neurons counted in the investigated DRG. In the left ganglia, the number of PNX-containing sensory neurons ranged from 7806 to 9963 per animal (9092 ± 1137) and accounted for 19.1 ± 0.6% of all the sensory neurons. Very similar results regarding both the number and percentage of PNX^+^ nerve cells were obtained in the right ganglia in which the number of PNX-labeled perikarya ranged from 8908 to 9712 per animal (9390 ± 425) and accounted for 19.3 ± 1.5% of all the sensory neurons.

The PNX^+^ sensory neurons were observed in all the selected C, Th, L and S DRG (20.2 ± 1.4%, 16.5 ± 1.5%, 19.4 ± 1.0% and 21.4 ± 2.5%, respectively). No statistically significant differences in the number of PNX-containing sensory nerve cells were observed between the left and right ganglia from the same level of the spinal cord and between the ganglia located in various levels of the spinal cord examined. Details concerning the relative frequency of PNX^+^ sensory neurons observed in the selected left and the right DRG taken from the C, Th, L and S levels of the spinal cord are shown in Table 1 and Figure 2.

#### 2.2.2. Morphometrical Characteristics of PNX^+^ Neurons

In the present study, the PNX^+^ DRG neurons belonged to all cell sizes: small (Sl) (neurons with an average diameter of up to 40 μm), medium (M) (neurons with a diameter from 41 μm to 70 μm) and large (Lg) (neurons with a diameter up to 71 μm). In general, the vast majority of PNX^+^ sensory neurons belonged to Sl-sized cells (73.9 ± 4.7%) which prevailed over the M-sized sensory neurons (24.9 ± 4.7%), while the Lg nerve cells (1.2 ± 0.8%) were only occasionally found in all the DRGs studied. There were no significant differences in the number of PNX-containing sensory neurons belonging to these three size classes between the left and the right DRGs. When comparing the relative frequencies of Sl, M and Lg perikarya in the left and right DRGs, Sl cells constituted the largest population of PNX^+^ neurons (70.6 ± 13.9% vs. 73.8 ± 4.9%) and M-sized cells constituted a smaller population (24.8 ± 5.3% vs. 25.1 ± 4.9%), while only a few Lg-sized cells (1.0 ± 0.9% vs. 1.1 ± 0.8%) were found. Moreover, no statistically significant differences in the number of PNX-labeled neurons belonging to the Sl-, M- and Lg-sized populations of sensory nerve cells were observed between the C, Th, L and S DRGs. The details on the relative frequency of PNX^+^ sensory neurons of the different size classes in the selected left and the right DRGs at the C, Th, L and S levels of the spinal cord are shown in Table 2 and Figure 3. Representative images of the PNX^+^ DRG nerve cells belonging to the different size classes are shown in Figure 4.

#### 2.2.3. Intraganglionic Distribution Patterns of PNX^+^ Neurons

To determine the intraganglionic distribution pattern(s) of the PNX^+^ sensory neurons in all the studied DRGs, the ‘mask’ shown in Figure 5 was applied to each analyzed ganglion section.

The PNX-containing sensory nerve cells were present in all five ganglion domains: central (Cn), peripheral (P), cranial (Cr), caudal (Cd) and middle (Md). In both the left and the right ganglia, the vast majority of observed nerve cells were clustered in the Cn part of the ganglion (29.1 ± 6.2% and 27.4 ± 2.2%, respectively), while fewer PNX^+^ sensory neurons (10.9 ± 3.6% and 12.1 ± 5.3%) were observed in the P domain of the DRG. The rest of the PNX^+^ perikarya were dispersed quite evenly in the Md (21.4 ± 3.1% and 21.5 ± 0.9%), Cr (19.7 ± 1.5% and 19.1 ± 2.2%) and Cd (18.9 ± 1.3% and 19.9 ± 1.2%) domains of the ganglion. There were no statistically significant differences in the number of PNX^+^ nerve cells observed in the same domains of the left and the right DRG studied. Moreover, a similar intraganglionic distribution pattern of PNX^+^ neurons was observed in all the studied DRGs from the C, Th, L and S segments of the spinal cord. The details concerning the intraganglionic distribution pattern of PNX^+^ sensory neurons in particular domains of the left and the right DRGs are shown in Table 3.

### 2.3. Immunohistochemical Characteristic of PNX-Containing Perikarya in DRGs 

A comprehensive analysis of double immunohistochemical staining showed the presence of PNX-IR sensory nerve cells together with other biologically active substances like CGRP, SP, nNOS, GAL, CRT, PACAP, CART or SOM in all the studied DRGs.

#### 2.3.1. PNX^+^/CGRP^+^ Nerve Cells

The largest subpopulation of PNX^+^ DRG sensory nerve cells were neurons simultaneously containing CGRP (Figure 6a,b). Such cells accounted for 96.1 ± 0.8% of the total population of PNX-IR neurons located in all the examined DRGs, while only a very small number of PNX^+^ nerve cells (3.9 ± 0.8%) did not simultaneously express CGRP. The PNX^+^/CGRP^+^ neurons were observed in all the studied DRGs of the C, Th, L and S segments of the spinal cord (96.9 ± 3.3%; 95.1 ± 4.0%; 96.9 ± 2.7%; and 96.1 ± 3.2%, respectively). Statistically significant differences in the number of PNX^+^/CGRP^+^ neurons were not observed between selected DRGs of a given level of the spinal cord nor between DRGs from different spinal cord levels.

Regarding their diameter, the PNX/CGRP-containing cells belonged to all three classes of afferent perikarya. The most numerous subpopulation of PNX^+^/CGRP^+^ neurons was that of Sl-sized cells (69.2 ± 8.8%); the M-sized neurons were found to be distinctly less numerous (29.1 ± 8.3%), while Lg PNX^+^ and CGRP-containing nerve cells (1.7 ± 0.5%) were rarely observed. No significant differences in the number of PNX^+^/CGRP^+^ DRG neurons belonging to the different size classes were observed between different levels of the spinal cord (Table 4).

The PNX^+^/CGRP^+^ sensory neurons were evenly dispersed in the individual DRG sections; a clear accumulation of these cells was observed in the Cn, Md, and Cr ganglion domains (25.3 ± 2.5%; 25.3 ± 0.7%; and 20.6 ± 1.3%, respectively), while a smaller number of PNX^+^/CGRP^+^ nerve cells was found in the Cd and the P parts (18.1 ± 1.8% and 10.7 ± 1.3%) of all the studied DRGs. The distribution pattern of the PNX^+^ and CGRP-containing sensory neurons in the different domains of the DRGs are shown in Figure 7a.

#### 2.3.2. PNX^+^/SP^+^ Nerve Cells

The second most numerous subpopulation of PNX^+^ sensory neurons in the investigated DRGs was that containing SP (88.5 ± 5.9%; Figure 6c,d), which was present in all the selected DRGs from the C, Th, L and S levels of the spinal cord (88.8 ± 4.9%, 87.5 ± 6.5%, 87.4 ± 7.3% and 90.3 ± 4.5%, respectively). No statistically significant differences were observed in the number of PNX^+^/SP^+^ sensory nerve cells between the selected DRGs in a given level of the spinal cord as well as between the different levels of the spinal cord.

In general, SP-IR was observed in all three size classes of PNX^+^ afferent perikarya, being present in 62.05 ± 6.7% of Sl, 36.9 ± 7.1% of M and 1.05 ± 0.4% of Lg cells. There were no significant differences in the number of PNX^+^/SP^+^ DRG neurons representing the different size classes between the different levels of the spinal cord (for details, see Table 5).

Additionally, most of the PNX^+^/SP^+^ neurons were unevenly distributed in different sensory ganglion domains. A clear accumulation of PNX^+^ and SP-containing sensory nerve cells was found in the Cn, Md, Cr and Cd domains of the DRGs (27.9 ± 4.6%, 22.2 ± 6.5%, 20.1 ± 2.1% and 18.9 ± 7.6%, respectively). A smaller number of PNX^+^/SP^+^ sensory neurons was found in the P part (10.9 ± 1.9%) of the studied DRGs. The distribution pattern of the PNX^+^ and SP-containing sensory neurons in the different domains of the DRGs are shown in Figure 7b.

#### 2.3.3. PNX^+^/nNOS^+^ Nerve Cells

The third largest subpopulation of PNX^+^ neurons in all the studied DRGs was that containing nNOS (52.1 ± 11.2%; Figure 6e,f). PNX^+^/nNOS^+^ sensory nerve cells were observed in all the studied DRGs from the C, Th, L and S levels of the spinal cord (50.1 ± 16.5%, 56.9 ± 11.6%, 49.8 ± 7.0% and 51.7 ± 9.5%, respectively). No statistically significant differences in the numbers of PNX^+^/nNOS^+^ neurons were observed between the selected DRGs of a given level of the spinal cord, nor between DRGs from different spinal cord levels.

In terms of their diameters, three classes of the PNX^+^/nNOS^+^ DRG neurons were observed. The most numerous subpopulation was formed by M-sized cells (52.4 ± 6.6%) and a less numerous one by Sl-sized perikarya (46.1 ± 6.9%), while Lg PNX^+^/nNOS^+^ neurons (1.5 ± 0.4%) were only observed sporadically. There were no significant differences in the number of PNX^+^/nNOS^+^ DRG neurons representing different size classes between the different levels of the spinal cord (for details, see Table 6).

The PNX^+^/nNOS^+^ sensory neurons were unevenly dispersed inside the ganglia, and were mainly clustered in the Md (25.6 ± 2.4%) and Cn (25.5 ± 1.4%) regions. The remaining nNOS-containing, PNX^+^ nerve cells were present in the Cr, Cd and P (19.7 ± 1.4%, 17.3 ± 2.9% and 11.9 ± 0.9%, respectively) domains of the studied DRGs. The distribution pattern of PNX^+^ and nNOS-containing sensory neurons in the different ganglionic domains is shown in Figure 7c.

#### 2.3.4. PNX^+^/GAL^+^ Nerve Cells

The PNX^+^/GAL-containing cells constituted one of the smaller populations of DRG cells (20.7 ± 5.7%; Figure 6g,h), and were observed in all the studied DRGs from the C, Th, L and S spinal cord levels (constituting 16.1 ± 5.3%, 18.7 ± 7.7%, 22.8 ± 4.9% and 25.5 ± 7.0% of all cells, respectively). No statistically significant differences in the number of PNX^+^/GAL^+^ neurons were observed between the selected DRGs of a given level of the spinal cord, nor between DRGs from different spinal cord levels.

Regarding their diameter, the PNX^+^/GAL^+^ DRG cells belonged to only two classes of afferent perikarya: the most numerous subpopulation of Sl-sized cells (99.1 ± 1.3%) and the distinctly less numerous subset of M-sized perikarya (0.9 ± 1.3%); PNX^+^/GAL^+^ Lg neurons were not found during the present study. There were no significant differences in the number of DRG neurons representing the different size classes between the different levels of the spinal cord (for details, see Table 7).

The largest accumulation of PNX^+^/GAL^+^ cells was observed in the Cn region of the studied DRGs (31.9 ± 4.1%), while the smallest number of these neurons was found in the P subdomain (9.3 ± 1.9%). The relative frequencies of the PNX^+^/GAL^+^ neurons were similar in the Md, Cr and Cd intraganglionic domains (21.6 ± 4.4%, 18.9 ± 0.7% and 18.3 ± 1.9%, respectively). The distribution pattern of PNX^+^ and GAL-containing sensory neurons in the different domains of the DRGs are shown in Figure 7d.

#### 2.3.5. PNX^+^/CRT^+^ Nerve Cells

A colocalization study of PNX with CRT unveiled the co-occurrence of these two biologically active substances in a small population (10.05 ± 1.6%; Figure 6i,j) of all the DRG cells studied. The PNX^+^/CRT^+^ sensory neurons were observed in all the selected DRGs from the C, Th, L and S levels of the spinal cord (constituting 11.2 ± 1.3%, 11.2 ± 4.9%, 10.5 ± 6.0% and 7.3 ± 1.5% of all cells, respectively). No statistically significant differences in the number of PNX^+^/CRT^+^ neurons were observed between the selected DRGs of a given level of the spinal cord, nor between DRGs from different spinal cord levels.

In terms of their diameter, three classes of PNX^+^/CRT^+^ DRG neurons were found. The vast majority of these cells were Sl-sized neurons (94.2 ± 6.2%). In addition, a small number of M-sized PNX^+^/CRT^+^ sensory nerve cells (5.1 ± 6.3%) were observed, while Lg cells (0.7 ± 0.8%) were only occasionally present in the examined DRGs. While there were no significant differences in the number of PNX^+^/CRT^+^ neurons belonging to different size classes in the DRGs of the C, Th and L segments of the spinal cord, large PNX^+^/CRT^+^ neurons were not been observed in S DRGs (for details, see Table 8).

The PNX^+^/CRT^+^ neurons were unevenly distributed between the different intraganglionic domains. The vast majority of PNX^+^ and CRT-containing sensory nerve cells were found in the Cn domain (33.4 ± 3.8%), while smaller numbers of PNX^+^/CRT^+^ neurons were found in the Md, Cr, Cd and P subdomains (18.6 ± 8.1%, 16.7 ± 0.7%, 16.5 ± 0.7% and 14.8 ± 1.4%, respectively). The distribution pattern of the PNX^+^ and CRT-containing sensory neurons in the different domains of the DRGs are shown in Figure 7e.

#### 2.3.6. PNX^+^/PACAP^+^ Nerve Cells

PACAP was found in 7.4 ± 0.8% (Figure 6k,l) of all PNX-containing neurons and it was present in all the selected DRGs at the C, Th, L and S levels of the spinal cord (7.5 ± 1.8%, 6.4 ± 0.6%, 7.4 ± 2.3% and 8.6 ± 2.2%, respectively). No statistically significant differences in the number of PNX^+^/PACAP^+^ neurons were observed between the selected DRGs of a given level of the spinal cord, nor between DRGs from different spinal cord levels.

In general, PACAP-IR was observed in all three size classes of PNX^+^ afferent perikarya, being present in 87.2 ± 4.5% of Sl, 12.1 ± 4.0% of M and 0.7 ± 0.6% of Lg cells. It should be stressed that statistically significant differences in the number of Sl- and M-sized PNX/PACAP-containing nerve cells were observed between the C and Th as well as the Th and S levels of the spinal cord. It should also be noted that the large PNX^+^/PACAP^+^ sensory neurons were only present in DRGs located in the C level of the spinal cord (see Table 9, Figure 8).

The PNX^+^/PACAP^+^ sensory neurons were dispersed throughout the ganglia, mainly in its Cn (26.5 ± 2.6%) and Md (21.0 ± 4.1%) regions. The remaining PACAP-containing, PNX^+^ nerve cells were present in the P, Cr and Cd domains of the studied DRGs (19.5 ± 2.2%, 18.3 ± 7.5% and 14.7 ± 5.1%, respectively). The distribution pattern of the PNX^+^ and PACAP-containing sensory neurons in the different domains of the DRGs is shown in Figure 7f.

#### 2.3.7. PNX^+^/CART^+^ Nerve Cells

Another small population of PNX^+^ DRG neurons contained CART (5.1 ± 4.1%; Figure 6m,n), and were present in all the selected DRGs from the C, Th, L and S levels of the spinal cord (5.4 ± 4,3%, 4.6 ± 3.6%, 4.9 ± 4.6% and 5.4 ± 5.0%, respectively). No statistically significant differences in the number of PNX^+^/CART^+^ neurons were observed between the selected DRGs of a given level of the spinal cord, nor between DRGs from different spinal cord levels.

In general, CART-IR was observed in all three size classes of afferent perikarya, being present in 55.1 ± 8.6% of M, 44.3 ± 8.6% of Sl and 0.6 ± 11.5% of Lg cells. No significant differences in the size of PNX^+^/CART^+^ DRG neurons between different levels of the spinal cord were observed (see Table 10).

Although the PNX^+^/CART^+^ sensory neurons were unevenly scattered throughout the individual ganglionic sections, a distinct accumulation of these cells was found in the Cn (31.9 ± 22.02%) ganglionic domains, while a smaller number of PNX^+^/CART^+^ sensory neurons was found in the Md, Cd, P and Cr parts (21.5 ± 12.1%, 18.1 ± 4.7%, 15.0 ± 6.4% and 13.5 ± 12.1%, respectively). The distribution pattern of the PNX^+^ and CART-containing sensory neurons in the different domains of the DRGs are shown in Figure 7g.

#### 2.3.8. PNX^+^/SOM^+^ Nerve Cells

The SOM-containing sensory neurons constituted the smallest subpopulation (4.7 ± 2.5%; Figure 6o,p) of all PNX^+^ perikarya, being observed in all the investigated C, Th, L and S DRGs (7.8 ± 6.0%, 6.3 ± 3.5%, 2.8 ± 1.5% and 2.1 ± 0.9%, respectively). No statistically significant differences in the number of PNX^+^/SOM^+^ neurons were observed between selected DRGs of a given level of the spinal cord, nor between DRGs from different spinal cord levels.

In terms of their diameter, three classes of the PNX^+^/SOM^+^ DRG neurons were observed. The most numerous subpopulation was Sl-sized cells (85.2 ± 9.4%) with a distinctly less numerous subset of M-sized perikarya (14.6 ± 5.3%), while the Lg PNX^+^/SOM^+^ neurons (0.2 ± 1.1%) were found sporadically in all the DRGs studied. There were no significant differences in the number of PNX^+^/SOM^+^ DRG neurons representing different size classes between the different levels of the spinal cord; however, it should be noted that Lg sensory neurons were not been observed in this subpopulation of PNX^+^ nerve cells in both the C and S ganglia of the spinal cord (for details, see Table 11).

A distinct accumulation of PNX-labeled and SOM^+^ cells was found in the Cn (28.2 ± 7.1%) and Md ganglionic regions (24.7 ± 2.5%), while these cells were less numerous in the Cr, P and Cd parts of the ganglia (17.0 ± 3.8%, 15.7 ± 8.1% and 14.4 ± 0.7%, respectively). The distribution pattern of the PNX^+^ and SOM-containing sensory neurons in the different domains of the DRGs are shown in Figure 7h.

## 3. Discussion

PNX is a newly identified peptide that, in the rat brain, is highly expressed in the hypothalamus, medial division of the central amygdaloid nucleus, the spinal trigeminal tract of the medulla and the spinocerebellar tract [11]. Moreover, enzyme immunoassays also detected a high level of PNX (>4.5 ng/g of tissue) in the spinal cords of rats [5]. Immunohistochemical studies in rat also revealed the presence of PNX in trigeminal ganglia, as well as in ganglia comprising neurons projecting to the superficial layers of the spinal cord, such as the DRGs and nodose ganglia [5]. These projections may belong to Aδ-terminals—thin, myelinated fibers capable of responding to both thermal and mechanical stimuli—or to C-fibers, which are unmyelinated, slow-conducting axons that can, due to their high activation threshold, detect painful stimuli. Both of these types of NFs are assigned to nociceptors as they respond to destructive, thermal, mechanical or chemical stimuli [12]. Furthermore, PNX^+^ afferent NFs targeting the spinal cord were revealed in mice and pigs and were present in all spinal cord segments at equal levels [6,13], whereas PNX was observed mainly in NFs located in the superficial DH layer, in laminae I and II [5,6]. It is also worth noting that PNX-IR NFs participate in skin innervation in rodents: subcutaneous injections of the fluorescent retrograde tracer Fluorogold labeled a population of DRG cells, some of which also contained PNX, thus raising the possibility that PNX released from these terminals could affect other sensory NF/sensory signaling pattern from the skin to the spinal cord [13].

In addition, the presence of “true” PNX was confirmed by the use of mass spectrometry (not only by showing that the major peak corresponds to the peptide, but also by verifying its presence using the fragmentation spectra of its individual amino acids) in the extracts of DRG cells (present study) as well as in the spinal cord (both rat [5] and mouse [13]).

The available literature implicates PNX in pain transmission, and it has been found that the intrathecal injection of amidated PNX suppresses visceral pain; however, it did not affect thermal pain sensation [5]. Moreover, PNX’s contribution to sensory modulation is also supported by its participation in inducing the itching effect, as revealed in a study by Cowan and colleagues [13].

To date, the presence of PNX in DRG neurons has only been studied in rodents [5,13]. Thus, for the first time, our study demonstrated the presence of PNX in afferent neurons of porcine DRGs. We have found that PNX^+^ neurons accounted for a relatively large subpopulation (approximately 20% of all cells) in all DRGs of each segment of the spinal cord. Furthermore, considering the differences in the size (diameter) of the PNX-containing cells, it was demonstrated that the vast majority of PNX-IR neurons were Sl-sized neurons (average diameter up to 40 μm). M-sized cells (40–70 μm) were distinctly less numerous, while Lg PNX^+^ cells (diameter > 71 μm) were only occasionally observed. This finding appears to be inconsistent with the previous results described in rodents, where many of the PNX^+^ neurons belonged to the M-sized DRG perikarya [5]. As there is a consensus that the size of the sensory cell is associated with different “physiological functional classes”, the data presented above may suggest a different involvement of PNX-positive cells in sensory modalities in rodents and the domestic pig. However, this hypothesis requires further verification.

The present study, for the first time, also focused on the immunohistochemical characteristics of PNX^+^ DRG neurons. We examined PNX colocalization patterns with several neurotransmitters/their markers that were previously revealed in DRG neurons, with particular attention to those biologically active molecules that were previously described in Sl- and/or M-diameter afferent cells: CGRP, SP, nNOS, GAL, CRT, PACAP, CART and SOM [14,15].

Previous studies revealed that, CGRP, together with SP (see below), appears to be the “canonical” transmission molecule of afferent neurons in all mammalian species studied so far (human, horse, dromedary camel, pig, cat, rat and mouse) [16,17,18,19,20]. According to the available data, the proportion of sensory neurons containing CGRP (or expressing mRNA encoding this peptide [21,22]) ranging, on average, between 50 and 60% of all DRG cells, regardless of the species studied. Furthermore, their axonal processes constitute the largest population of nerve fibers forming not only individual laminae of the dorsal horns of the spinal cord (especially laminae I-III, V and X), but also Lissauer’s tract [17]. Interestingly, this pattern of intraspinal distribution of CGRP-positive central projections of DRG cells persists in all spinal cord neuromeres, which is most prominent in the thoracic ones [16].

In addition to being involved in the conduction of sensory and pain stimuli in a wide variety of mammals, including human [17,18,23,24,25], CGRP is also known to be a very potent vasodilator [26] and a deficiency in CGRP release is related to a lack of a vasodilation reflex [27]; thus, blockade of CGRP secretion exerts an analgesic effect in people suffering from migraine [28].

However, CGRP, along with SP, released from the terminals of DRG neurons in response to noxious mechanical, chemical or thermal stimuli [29] may also contribute to the development and maintenance of neurogenic inflammation: CGRP acts as an extremely potent vasodilator [30] while SP, acting on neurokinin 1 (NK1) receptors, evokes increased vascular permeability [31]. Therefore, it can be concluded that the intensified secretion of these peptides may contribute to the formation of edema, increased blood flow and the influx of inflammatory cells at the site of inflammation.

The above-mentioned close interdependence and cooperation of CGRP and SP in numerous regulatory loops seem to result from the fact that both substances coexist very often in the same sensor cell [14,17,20] from which they are released together and synergistically interact with target cells, supporting each other. For example, as revealed by Biella and co-workers [32], the excitatory effects of SP arising from spontaneous and noxious activity were significantly enhanced by CGRP in rats. Although the mechanism by which CGRP may potentiate the effects of SP is not fully clear, there is evidence that CGRP may delay the enzymatic degradation of SP [33,34]. CGRP has also been reported to increase the release of SP [35] as well as the excitatory amino acid transmitters glutamate and aspartate [36] from central terminals of DRG neurons, possibly leading to the strengthening of synaptic connections in the spinal dorsal horn, as well as increasing the effectiveness of these substances on the peripheral targets (for details, see [37]).

The second largest population of PNX^+^ DRG cells observed in this study were SP-containing neurons, accounting for approximately 89% of all PNX^+^ afferent cells. This observation seems to be in stark contrast to the data obtained in studies of the co-occurrence of PNX and SP in mouse DRG cells, where both neurotransmitters were present in separate cell subpopulations [5]. Therefore, it seems reasonable to hypothesize that PNX is probably involved in regulating different functions of DRG sensory cells in both species.

As shortly mentioned above, SP, like CGRP, is primarily considered a transmitter of sensory and pain stimuli, the presence of which has been shown in DRG cells in every species studied so far, including humans [38,39,40,41]. So far, studies on the presence and distribution patterns of SP^+^ cells in the DRGs of the domestic pig have shown the presence of cells containing this tachykinin, especially in subpopulations of Sl- and M-sized neurons. Moreover, there is irrefutable evidence that SP occurs in both the somatic as well as visceral subclasses of these perikarya [14,42]. This is consistent with the data obtained in other species studied so far, including humans [21], guinea pigs [39], mice [5] and rats [43,44], with regard to both the number of chemically encoded afferent cells and their classification into individual size classes.

While the co-existence of PNX with CGRP in the studied porcine DRG cells was shown in a very high percentage of cells (approximately 96% of all PNX^+^ cells), only slightly more than half of the CGRP^+^ cells (56%) also contained PNX. Moreover, a very similar picture emerged from the studies on the co-occurrence pattern of PNX and SP performed in this study: almost 90% (exactly 88.5%) of all PNX^+^ cells contained SP, and approximately 50% of all SP^+^ cells were also PNX-positive. The co-localization of both peptides was observed mainly in Sl- and M-sized ganglion cells.

Thus, based on the comparison of the relative numbers of DRG cells containing the individual transmitters, there is no mathematical possibility that all three peptides do not co-occur in at least approximately 80–85% of all PNX^+^ cells. This suggestion is further, although indirectly, supported by the observation of the high degree of co-occurrence of PNX with SP and/or CGRP in the axonal fibers of DRG cells forming the dorsal horn laminae of the porcine spinal cord [6].

Moreover, considering the degree of co-localization of PNX and nNOS in DRG cells (approximately 50% of all PNX^+^ cells; see below), it seems that neurons containing all four transmitting substances must also be present in at least 30% of all the porcine DRG cells. Unfortunately, at present, insufficient knowledge of the possible functions of PNX in peripheral sensory pathways prevents us from suggesting its physiological significance, either in the case of hypothetical PNX^+^/CGRP^+^/SP^+^ cells or in the population of cells additionally co-expressing nNOS.

nNOS is a marker of the nitrergic subpopulation of afferent cells that are involved in the regulation of numerous physiological processes in the peripheral and central nervous systems. Nitric oxide (NO), produced by nNOS, is an important neurotransmitter in the periphery, mainly in the control of, among others, neuronal plasticity [45], neurogenesis [46], neuroprotection [47] as well as modulation of nociceptive transmission in neuropathic pain [48] and it is believed that inhibition of nNOS may be a valuable strategy in the treatment of not only neuropathic pain [49] but also several other illnesses [50].

In the present study, PNX^+^/nNOS-containing neurons formed the third largest (approximately 50%) population among all afferent cells capable of synthesizing and releasing PNX as their transmitter; the vast majority of these neurons were Sl- and M-sized cells. This is well in line with the data obtained in other species (humans, rats, mice, sheep, dromedary camels [20,51,52,53]) where nitrergic afferent neurons were found in an average of 40–50% of all DRG cells, with the highest percentage in the subpopulation of Sl- and M-sized neurons. It should be emphasized that the high degree of nNOS co-localization with SP and CGRP, reported by Russo and colleagues [53], additionally supports the above hypothesis, suggesting the existence of a PNX^+^/SP^+^/CGRP^+^/NOS^+^ population of DRG cells.

The co-existence of PNX and GAL was found exclusively in porcine DRGs in approximately one-fifth (20.7%) of Sl-sized neurons. As neither the immunoreactivity to GAL alone nor the co-occurrence of PNX and GAL immunoreactivities was observed in M- or Lg-sized cells, this strongly suggests the involvement of cells containing these two peptides in the regulation of nociceptive activities in the domestic pig. This suggestion corresponds well with the data from studies in other species, especially rats: GAL was also found in Sl-sized DRG cells in rodents (in a few cells [54]) and humans (approximately 12% [22]); moreover, it has also been observed that their number increases dramatically after peripheral nerve damage [55,56]. Numerous studies indicate that GAL may reduce the perception and transmission of pain by increasing the pain threshold as a result of increasing K^+^ conductance or decreasing Ca^2+^ conductance in the brain, spinal cord and peripheral neurons [57,58]. Moreover, in behavioral studies, it was found that reducing the effect of GAL leads to an increase in neuropathic pain-like behaviors and, on the contrary, the upregulation of GAL leads to a decrease in neuropathic pain in animal models [56].

In contrast to its co-occurrence pattern with CGRP, SP or GAL, PNX co-localized with the other studied substances (neuropeptides: PACAP, CART and SOM; calcium-binding protein: CRT) in only a relatively few porcine DRG cells.

In the case of the co-occurrence of PNX and PACAP, a significantly lower degree of colocalization was observed in this study (about 7% of all PNX^+^ cells simultaneously contained PACAP); at the same time, however, it should be emphasized that the vast majority of neurons chemically coded by the presence of both peptides belonged to the population of small, probably nociceptive, neurons. These observations agree fairly well with the data obtained in rodents, where PACAP^+^ cells were part of a Sl-diameter neuronal population [59,60]. It should be noted, however, that the number of PACAP^+^ cells in rat DRGs was slightly higher, reaching approximately 10–17.5% of all sensory nerve cells [61], suggesting interspecies differences.

It is worth noting that both PACAP and GAL are dramatically up-regulated in damaged DRG cells (up to 75% in the rat [60]), especially in Sl-sized cells (probably nociceptive perikarya), and act as neuroprotective and/or pro-regenerative peptides in this scenario [62,63,64]. Unfortunately, so far, there are no data on the possible functions of PNX in these regulatory mechanisms, and therefore the question of the possible existence of a subpopulation of PNX^+^/GAL^+^/PACAP^+^ cells, as well as their presumed targets and possible regulatory functions remains open and requires further, more detailed research.

In contrast to the patterns of PNX co-localization with the other biologically active agents discussed above, where the vast majority of observed PNX^+^ cells belonged almost exclusively to Sl- or M-diameter cells, PNX and CART co-localization was observed in the pig in all three neuronal size classes, with the M-sized PNX^+^/CART^+^ cells being the most numerous. While the data on the presence of CART in porcine DRG cells collected in this study appear to be in full agreement with the results obtained by Zacharko-Siembida and colleagues [65] in the lumbar porcine DRGs (in both studies, the relative number of CART+ cells oscillated around 5% of all DRG cells), a slightly more numerous (approximately 10%) population was observed in rodents [66]. Remarkably, Zacharko-Siembida and colleagues found that the vast majority (approximately 70%) of porcine CART^+^ DRG cells simultaneously contain CGRP. This allows for the assumption that some of the sensory cells may also contain, in addition to PNX^+^/CGRP^+^/SP^+^/NOS^+^, CART, which additionally suggests the need for in-depth research on both the chemistry and the functional significance of these sensory cells.

The last of the neuropeptides whose pattern of co-occurrence with PNX was the subject of interest in this work was SOM, a substance present in both the pig and the rat [67] in a small (up to 10%) subpopulation of the DRG cells. It has been shown that a small fraction (about 6% of all PNX^+^ cells) also contains SOM, which, in the light of the available literature data (both PNX and SOM are implicated as transmitters subserving the itch conduction pathway [13,68,69]) the most recent data, see [70], suggests that these cells are representatives of pruriceptors. However, further studies are needed to unveil the exact relevance of PNX in this neural modality.

Intracellular calcium-binding proteins may play an important role in the nociceptive excitability of neurons [71,72], as well as in proprioception [73], as evidenced by the presence of calbindin (CB) in excitatory interneurons within the superficial DH [74]. Taking this into account, we decided to analyze the co-occurrence of CRT (protein closely related to the above-mentioned CB) with PNX in porcine primary afferent cells. It was found that PNX coexisted with CRT in a relatively small population of DRG neurons (approximately 10.5% of all PNX-containing cells), due to the fact that CRT was expressed mainly in Lg-sized neurons, and the vast majority of PNX^+^ neurons were small-sized cells. These data are in line with reports emerging from studies concerning the CB or CRT expression patterns in rats and fowl [15,73,75], which showed that the vast majority of CB- or CRT-positive neurons were M to Lg in size [73].

### Limitations of the Study

The most important limitation of the present study was the inability to perform simultaneous colocalization studies of a larger number of neurotransmitters in the same nerve cell, due to the unavailability of appropriate primary and/or secondary antibodies. Similarly, the authors are aware that the value of this work would be higher if it were possible to demonstrate expression patterns of membrane/cytosolic receptor subtypes specific to the studied neurotransmitters; unfortunately, as in the case of studies based on multiple colocalizations in a single afferent cell, the unavailability of antibodies recognizing the mentioned receptors was an obstacle. The last limitation was the inability to link the observed PNX^+^ cell subtypes with their target tissues within the assumptions of this study: an analysis of the presence of PNX^+^ cells in the DRGs of so many spinal cord neuromeres would require the injection of a retrograde neuronal tracer into individual internal organs in such a large number that the planned experiment was simply becoming impossible. On the other hand, the use of an anterograde tracing technique was excluded due to the need to inject the tracer directly into the studied DRG, which would certainly cause changes in the expression pattern(s) of the tested substances.

## 4. Materials and Methods

### 4.1. Experimental Animals

The investigations were performed on 6 juvenile female pigs (8–12 weeks old, 15–20 kg body weight—b.w.) of the Large White Polish breed. The animals were kept under standard laboratory conditions. They were fed standard fodder (Grower Plus, Wipasz, Wadąg, Poland) and had free access to water. All surgical procedures were performed following the rules of the local Ethics Commission (affiliated with the National Ethics Commission for Animal Experimentation, Polish Ministry of Science and Higher Education; decision No. 40/2020 from 22 July 2020).

### 4.2. Surgical Procedures

All animals were pretreated with atropine (Polfa, Warsaw, Poland; 0.05 mg/kg b.w., subcutaneous (s.c.) injection) and azaperone (Stresnil, Janssen Pharmaceutica, Belgium; 2.5 mg/kg b.w., intramuscular (i.m) injection) and after thirty minutes, they were deeply anesthetized with sodium pentobarbital (Tiopental, Sandoz, Poland; 0.5 g per animal, administered according to the effect). The 5th lumbar DRGs (L5) were collected intraoperatively from 3 out of 6 pigs, and they were used for PNX identification using mass spectroscopy (MS). Next, the anesthesia was deepened, and after cessation of breathing, all the animals were transcardially perfused with freshly prepared 4% paraformaldehyde in 0.1 M phosphate buffer (pH 7.4). Whole spinal cords, including the attached DRGs, were gently dissected, sectioned into individual neuromeres (assuming the anatomical extent of their *fila radicularia* as boundaries) and each neuromere was fixed in the same fixative (10 min at room temperature), washed several times in 0.1 M phosphate buffer (pH 7.4; 4 °C; twice daily for three days) and stored in 18% buffered sucrose at 4 °C (for two weeks) until sectioned.

### 4.3. Protein Extraction

The proteomics analysis of the DRGs was performed at the Mass Spectrometry Laboratory at the Institute of Biochemistry and Biophysics of the Polish Academy of Sciences. The tissue was transferred to 2 mL tubes containing a mix of 1.4 mm, 2.8 mm and 5.0 mm zirconium oxide beads. The lysis was performed in 500 μL of 25% trifluoroethanol (TFE) in 100 mM triethylammonium bicarbonate buffer (TEAB). The tissue was lysed using a Precellys Evolution homogenizer (Bertin Technologies, Montigny-le-Bretonneux, France) using 6 rounds of a program with the following parameters: 10 cycles of 6800 rpm for 60 s with a 20 s pause in between. Homogenization was performed at 4 °C with dry ice cooling. After lysis, the samples were centrifuged (14,000× *g*, 30 min, 4 °C) and transferred to 1.5 mL tubes with low retention. Cysteine residues were reduced by incubation with 10 mM Tris (2-carboxyethyl)phosphine (TCEP) for 60 min, followed by blocking with 25 mM chloroacetamide. A 100 μL volume of the lysate was diluted to 500 μL with a 10% aqueous solution of acetonitrile (ACN), vortexed and briefly spun to remove the precipitated proteins. The sample was transferred to a 30 kDa cut-off filter (Vivacon 500, Sartorius, Göttingen, Germany). After centrifugation (14,000× *g*, 2 h, 25 °C), the filtrate was transferred to a new Eppendorf tube, dried in a SpeedVac and reconstituted in 40 μL of 0.1% formic acid.

### 4.4. Mass Spectroscopy (MS)

The samples were analyzed using a liquid chromatography–mass spectrometry (LC-MS) system composed of Evosep One (Evosep Biosystems, Odense, Denmark) coupled to an Orbitrap Exploris 480 mass spectrometer (Thermo Fisher Scientific, Bremen, Germany). A 20 μL volume of the sample was loaded onto disposable Evotips C18 trap columns as described previously [11]. Chromatography was carried out at a flow rate of 250 nl/min using the 88 min (15 samples per day) preformed gradient on N EV1106 analytical column (Dr Maisch C18 AQ, 1.9 μm beads, 150 μm ID, 15 cm long; Evosep Biosystems, Odense, Denmark). Data were acquired in positive mode with a data-dependent method using the following parameters: MS1 resolution was set at 120 000 with a 500% normalized automatic gain control (AGC) target, auto maximum inject time and a scan range of 250 to 1200 *m*/*z*; cycle time was set to 1 s. The ions with an *m*/*z* corresponding to the PNX protein with amidation within 1.6 *m*/*z* window were subjected to fragmentation with a normalized collision energy of 30%. For MS2, the resolution was set at 30,000 with a 1000% normalized AGC target and auto maximum inject time. The spray voltage was set at 2.1 kV, the funnel radio frequency (RF) level at 40, and the heated capillary temperature at 275 °C.

### 4.5. MS Data Analysis

Protein identification was performed using the Proteome Discoverer software suite (version 2.4.0.305) and the Sus scrofa full Uniprot database (version 2022_02). The search included non-cleaved proteins, Carboamidomethylation (C) was set as a fixed modification and Oxidation (M) and Amidation on Protein C-term were set as a variable ones. Fixed Value PSM (peptide-spectra match) Validator was used for PSM validation. At the consensus step, the basic Proteome Discoverer settings were used.

### 4.6. Sectioning of the Tissue Samples

Three representative DRGs from each level of the spinal cord were selected for this study. Bilateral DRGs from cervical (C: C1, C4, C7), thoracic (Th: Th1, Th7, Th15), lumbar (L: L1, L3, L6) and sacral (S: S1, S3) neuromeres were located pairwise on a pre-cooled block of the optimal cutting temperature compound medium (OCT) in a manner that allowed for the easy determination of the position of individual subdomains of the DRG sections under a microscope. Frozen ganglia were cut with an HM525 Zeiss freezing microtome into transverse 10 μm thick serial sections (four sections on each slide) and mounted on chrome alum–gelatine-coated slides.

### 4.7. Immunohistochemical Procedure and Estimation of the Total Number of PNX-Containing DRG Neurons

Serial 10 μm thick DRG sections were processed for double-labeling immunofluorescence (following a previously described protocol [14]) using combinations of primary immunosera raised in different species. After immersion of the tissues in a blocking solution containing 0.1% bovine serum albumin, 1% Triton X 100, 0.01% sodium azide (NaN_3_), 0.05% thimerosal and 10% normal goat serum in 0.01 M phosphate-buffered saline (PBS) for 1 h at room temperature to reduce non-specific background staining, the sections were repeatedly rinsed in PBS, and then incubated overnight at room temperature with PNX antiserum applied in a mixture with antisera against: CART, CGRP, CRT, GAL, nNOS, PACAP, SOM and SP. A combination of antisera against PNX and protein gene product 9.5 (PGP 9.5) was used to accurately determine the distribution pattern and the total number of PNX cells in the right and left DRGs from the different levels of the spinal cord (C, Th, L and S). Primary antisera were visualized by fluorescein isothiocyanate (FITC)-conjugated rat, guinea pig or mouse immunoglobulin G (IgG)-specific secondary antisera or streptavidin (CY3) rabbit-specific antiserum (Table 12). After rinsing the tissue with PBS, Fluoromount™ Aqueous Mounting Medium (Sigma Aldrich, Saint Louis, MO, USA, F4680) was applied to the section and then the tissue was covered with a coverslip.

PNX/PGP 9.5-positive perikarya were counted in every fourth section of six slides per DRG studied, while the colocalization of PNX with other biologically active substances was assessed from three slides (one section on the given slide for colocalization experiment) taken symmetrically from each one-third (anterior, middle, posterior; total = 9 slides) of each DRG studied: C (C1, C4, C7), Th (Th1, Th7, Th15), L (L1, L3, L6) and S (S1, S3). While the number of DRG cells belonging to specific neuronal subpopulations was manually counted within the entire section of the ganglion being assessed (only neuronal profiles in which the cell nucleus was clearly visible were counted), the diameters of PNX-positive (PNX^+^) perikarya were measured using a subroutine of the image analysis software (AnalySIS version 3.0, Soft Imaging System GmbH, Germany). The neurons were divided into three size classes: Sl (average diameter up to 40 μm), M (diameter 41–70 μm) and Lg (diameter ≥ 71 μm). To determine the relative frequency of the appropriately chemically coded subpopulation of the studied sensory neurons, all profiles of cells containing, for example, CGRP (visualized with FITC-labeled secondary antibody) and PNX (visualized with CY3-labeled secondary biotinylated antibody) were counted within the examined DRG section. The correctness of the assessment was confirmed using a special filter allowing the simultaneous observation of FITC and CY3 excitation. The combinations of primary antibodies employed in this study for double labeling are presented in Table 13.

The labeled sections were viewed using an Olympus BX61 microscope (Olympus, Hamburg, Germany) equipped with an appropriate filter set for CY3 and FITC. Micrographs were taken using an Olympus XM10 digital camera (Tokyo, Japan). The microscope was equipped with cellSens Dimension 1.7 Image Processing software (Olympus Soft Imaging Solutions, Münster, Germany). Finally, the obtained data were pooled for each animal and are presented as mean ± SD and were statistically analyzed using the Student’s two-tailed *t*-test for unpaired data and one-way ANOVA with mean values compared using Tukey’s multiple comparison test. The results were statistically analyzed using GraphPad PRISM 8.0 software (GraphPad Software, La Jolla, CA, USA). The differences were considered to be significant at *p* < 0.05.

### 4.8. Control of Specificity of Immunohistochemical Procedures

Standard controls, i.e., preabsorption for the neuropeptide antisera (20 μg of appropriate antigen per 1 mL of the corresponding antibody at working dilution; all antigens were purchased from Peninsula (San Carlos, CA, USA), R&D Systems (Wiesbaden, Germany), SWANT (Burgdorf, Switzerland), Sigma, Phoenix (Burlingame, CA, USA)—Table 14), as well as omission and replacement of the respective primary antiserum with the corresponding non-immune serum completely abolished immunofluorescence and eliminated the specific staining. Photos showing the preabsorption controls are presented in Figure 9.

## 5. Conclusions

In summary, our research is the second work to show the location of PNX in DRGs, and the first performed in a large mammal (pig). Moreover, we have revealed the coexistence pattern(s) of PNX with a plethora of other neurotransmitters in the examined nerve cells, suggesting the existence of multiple transmitters using DRG cells (i.e., PNX/CGRP/SP/NOS/CART), that may, most probably, be involved in the control of blood vessel tone, maintenance/regulation of neurogenic inflammation process, as well as in the conveyance of itching sensations. These data may be helpful in better understanding the functional role of PNX in transmitting various types of sensory information and its possible impact on the functioning of many organs in the body. Additionally, the presented data indicate that PNX-IR appears in a heterogeneous group of sensory ganglion cells, from small to large, suggesting that this peptide can be subordinated to different sensory modalities.

## Figures and Tables

**Figure 1 ijms-24-16647-f001:**
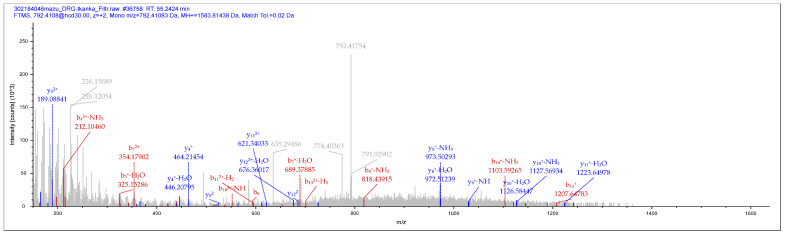
Phoenixin-14 (PNX) fragmentation spectrum-amidated (-0.98402 Da), derived from dorsal root ganglion (DRG) L5 from pig number 1 (DRG1).

**Figure 2 ijms-24-16647-f002:**
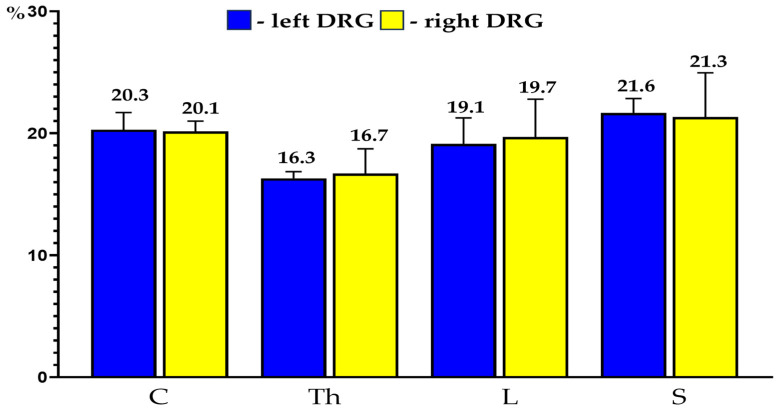
Bar diagram showing the relative percentages of PNX^+^ sensory neurons located in the left (blue bars) and the right (yellow bars) DRGs of the C, Th, L and S segments of the spinal cord. The obtained data were pooled for each animal and are presented as mean ± SD (N = 6 animals).

**Figure 3 ijms-24-16647-f003:**
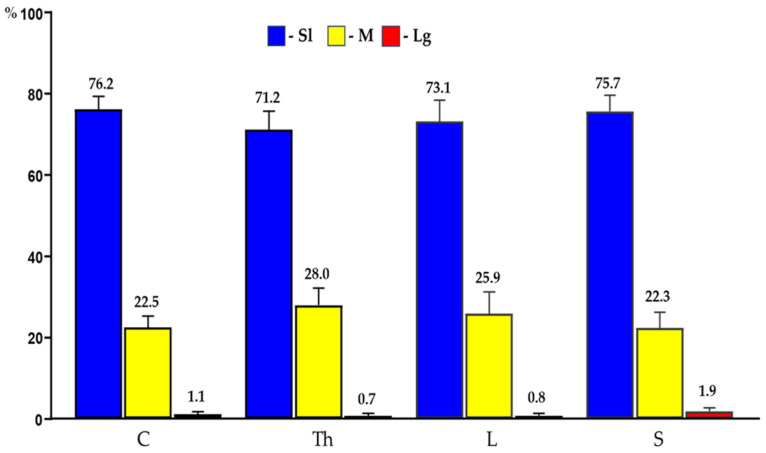
Bar diagram summarizing the percentages of differently sized (Sl; M; Lg) PNX^+^ sensory nerve cells found in C, Th, L and S segments of the spinal cord. The obtained data were pooled for each animal and are presented as mean ± SD (N = 6 animals).

**Figure 4 ijms-24-16647-f004:**
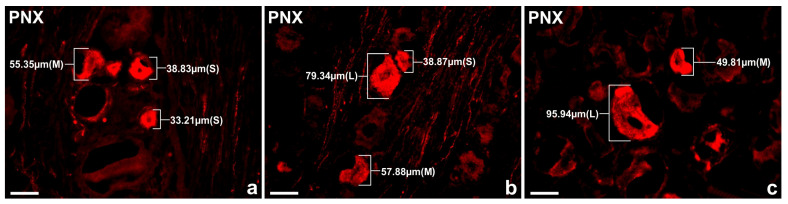
Representative images of the PNX^+^ nerve cells in the porcine DRG L1 belonging to the different size classes: (**a**) Sl cells (33.21 μm, 38.83 μm), M cells (55.35 μm); (**b**) Sl cell (38.87 μm), M cell (57.88 μm), Lg cell (79.34 μm); (**c**) M cell (49.81 μm), Lg cell (95.94 μm). Scale bar: 50 μm.

**Figure 5 ijms-24-16647-f005:**
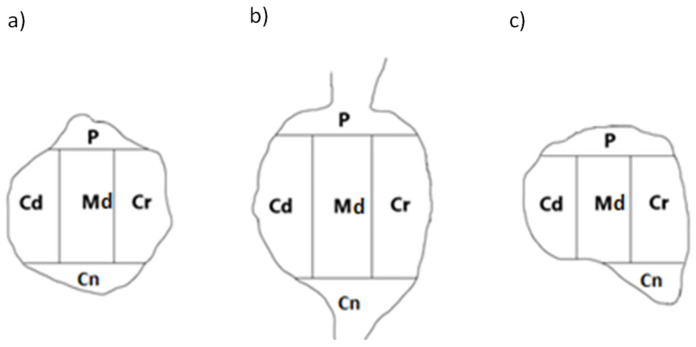
A schematic diagram of the DRG section showing its arbitrary division into topographical domains and in which, the occurrence and relative frequency of PNX-containing sensory neurons were studied: P—peripheral domain; Cr—cranial domain; Cd—caudal domain; Cn—central domain of the DRG; Md—middle ganglion area. Section from the (**a**) proximal (**b**) middle and (**c**) distal part of the ganglion.

**Figure 6 ijms-24-16647-f006:**
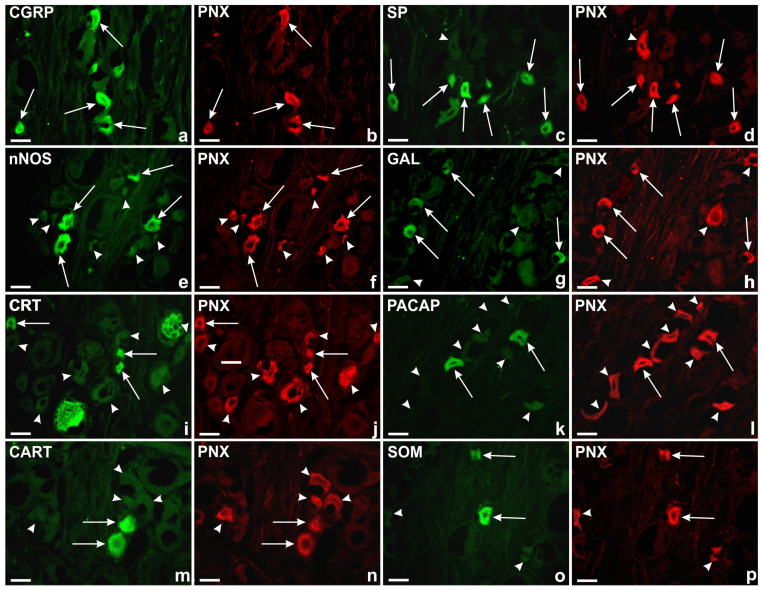
Representative images of the DRG sensory neurons. All the images were taken separately in green (fluorescein isothiocyanate—FITC) (**a**,**c**,**e**,**g**,**i**,**k**,**m**,**o**) and red (streptavidin—CY3) (**b**,**d**,**f**,**h**,**j**,**l**,**n**,**p**) fluorescent channels. Long arrows represent PNX^+^ cells (**b**,**d**,**f**,**h**,**j**,**l**,**n**,**p**) that simultaneously contain (**a**) calcitonin gene-related peptide (CGRP; 96.1%), (**c**) substance P (SP; 88.5%), (**e**) neuronal nitric oxide synthase (nNOS; 52.1%), (**g**) galanin (GAL; 20.7%), (**i**) calretinin (CRT; 10%), (**k**) pituitary adenylate cyclase-activating polypeptide (PACAP; 7.4%), (**m**) cocaine- and amphetamine-regulated transcript (CART; 5.1%), (**o**) somatostatin (SOM; 4.7%); arrowheads represent DRG PNX^+^ sensory nerve cells (**d**,**f**,**h**,**j**,**l**,**n**,**p**) which were simultaneously immunonegative for CGRP (**a**), SP (**c**), nNOS (**e**), GAL (**g**), CRT (**i**), PACAP (**k**), CART (**m**) and SOM (**o**). Scale bar: 50 μm.

**Figure 7 ijms-24-16647-f007:**
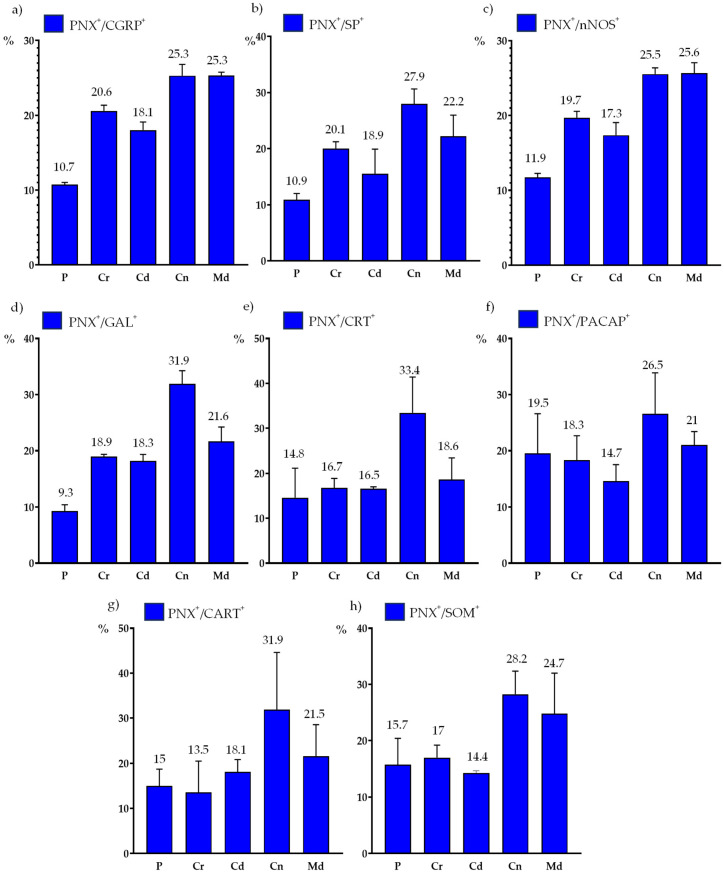
Distribution pattern of PNX^+^ neurons containing neurotransmitters in different domains (P, Cr, Cd, Cn and Md) of the DRG: (**a**) PNX^+^/CGRP^+^ sensory neurons, (**b**) PNX^+^/SP^+^ sensory neurons, (**c**) PNX^+^/nNOS^+^ sensory neurons, (**d**) PNX^+^/GAL^+^ sensory neurons, (**e**) PNX^+^/CRT^+^ sensory neurons, (**f**) PNX^+^/PACAP^+^ sensory neurons, (**g**) PNX^+^/CART^+^ sensory neurons, (**h**) PNX^+^/SOM^+^ sensory neurons. The obtained data were pooled for each animal and are presented as mean ± SD (N = 6 animals).

**Figure 8 ijms-24-16647-f008:**
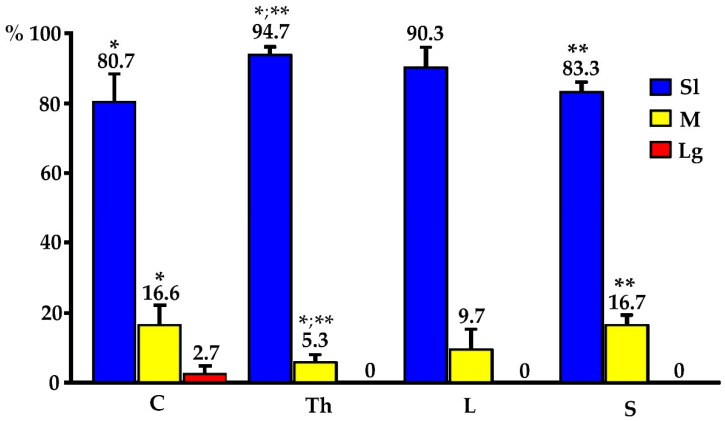
Relative frequency of different sizes of PNX^+^/PACAP^+^ perikarya found in DRGs of the C, Th, L and S spinal cord segments. The obtained data were pooled for each animal and are presented as mean ± SD (N = 6 animals); the data were statistically analyzed using the Student’s two-tailed *t*-test for unpaired data and using one-way ANOVA with mean values compared using Tukey’s multiple comparison test. Asterisks mark statistically significant differences: * *p* < 0.05, ** *p* = 0.005. The results were statistically analyzed using GraphPad PRISM 8.0 software.

**Figure 9 ijms-24-16647-f009:**
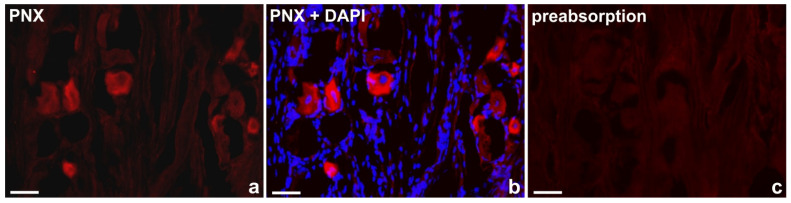
Representative examples of the immunofluorescence staining specificity control procedures used in the study: (**a**) “classical” staining with anti-PNX antibody, visualized by CY3, (**b**) anti-PNX antibody + 4′,6-diamidino-2-phenylindole (DAPI) as a counterstain, (**c**) antiserum anti-PNX antibody preabsorbed with PNX amide. Scale bar: 50 μm.

**Table 1 ijms-24-16647-t001:** Relative frequency of PNX-positive (PNX^+^) neuronal profiles located in the left and right DRGs of the cervical (C), thoracic (Th), lumbar (L) and sacral (S) levels of the spinal cord. The obtained data were pooled for each animal and are presented as mean ± standard deviation (SD) (N = 6 animals).

Segments of the Spinal Cord (%)											
DRG	C1	C4	C7	Th1	Th7	Th15	L1	L3	L6	S1	S3
left	21.5 ± 3.9	17.5 ± 3.1	21.9 ± 1.7	16.2 ± 2.0	17.3 ± 0.7	15.4 ± 2.3	17.8 ± 2.4	16.3 ± 0.8	23.3 ± 3.6	22.8 ± 2.0	20.4 ± 4.6
right	18.5 ± 7.2	21.1 ± 6.6	20.9 ± 1.0	16.8 ± 1.9	17.2 ± 0.6	16.1 ± 3.7	15.7± 4.1	17.6 ± 2.0	25.8 ± 1.2	24.6 ± 0.9	18.0 ± 7.6

**Table 2 ijms-24-16647-t002:** Percentages of PNX^+^ neuronal populations of different sizes (small—Sl; medium—M; large—Lg) located in the left and the right DRGs of the C, Th, L and S segments of the spinal cord. The obtained data were pooled for each animal and are presented as mean ± SD (N = 6 animals).

Segments of The Spinal Cord (%)												
	C			Th			L			S		
DRG	Sl	M	Lg	Sl	M	Lg	Sl	M	Lg	Sl	M	Lg
left	74.7 ± 3.1	23.9 ± 2.3	1.4 ± 0.9	70.6 ± 3.1	28.5 ± 2.9	0.9 ± 0.3	75.4 ± 5.9	23.8 ± 5.8	0.8 ± 0.6	76.3 ± 1.0	21.9 ± 0.8	1.8 ± 0.2
right	76.7 ± 3.1	22.1 ± 2.9	1.2 ± 0.6	71.4 ± 4.9	27.8 ± 4.7	0.8 ± 0.7	72.4 ± 5.2	26.6 ± 5.2	1.0 ± 0.6	75.4 ± 4.8	22.5 ± 4.8	2.1 ± 0.9

**Table 3 ijms-24-16647-t003:** Intraganglionic distribution pattern of PNX^+^ neuronal populations located in the bilateral DRGs in the C, Th, L and S segments of the spinal cord. The obtained data were pooled for each animal and are presented as mean ± SD (N = 6 animals).

Segments of the Spinal Cord (%)								
DRG Subdomain	C		Th		L		S	
	Left	Right	Left	Right	Left	Right	Left	Right
P	15.1 ± 2.7	19.4 ± 2.1	12.1 ± 2.3	11.2 ± 1.8	10.7 ± 0.7	11.3 ± 2.0	11.3 ± 2.1	10.6 ± 1.3
Cr	20.9 ± 1.7	18.7 ± 1.2	20.4 ± 1.7	22.1 ± 1.6	20.1 ± 1.4	18.5 ± 3.8	20.4 ± 1.9	21.3 ± 4.2
Cd	17.3 ± 3.2	16.6 ± 1.5	17.4 ± 1.9	19.1 ± 2.4	19.2 ± 3.5	19.5 ± 1.9	18.1 ± 1.4	19.4 ± 4.1
Cn	24.2 ± 5.0	25.1 ± 3.9	26.1 ± 4.5	26.3 ± 1.8	27.9 ± 5.7	28.8 ± 4.5	31.1 ± 6.0	26.1 ± 6.7
Md	22.5 ± 1.5	20.2 ± 2.1	24.0 ± 1.3	21.3 ± 0.9	22.1 ± 5.8	21.9 ± 2.1	19.1 ± 1.9	22.6 ± 4.4

**Table 4 ijms-24-16647-t004:** Relative frequency of different sizes (Sl, M, Lg) of PNX^+^/CGRP^+^ perikarya found in DRGs of the C, Th, L and S spinal cord segments. The obtained data were pooled for each animal and are presented as mean ± SD (N = 6 animals).

	Segments of the Spinal Cord (%)			
Size of DRG Cells	C	Th	L	S
Sl	57.1 ± 21.4	71.5 ± 4.9	73.3 ± 11.4	75.1 ± 6.7
M	42.10 ± 21.7	27.0 ± 4.9	26.1 ± 11.9	21.3 ± 3.2
Lg	0.8 ± 0.7	1.5 ± 0.6	0.6 ± 0.6	3.6 ± 3.6

**Table 5 ijms-24-16647-t005:** Relative frequency of different sizes of PNX^+^/SP^+^ perikarya found in DRGs of the C, Th, L and S spinal cord segments. The obtained data were pooled for each animal and are presented as mean ± SD (N = 6 animals).

	Segments of the Spinal Cord (%)			
Size of DRG Cells	C	Th	L	S
Sl	53.4 ± 25.1	59.8 ± 11.2	67.3 ± 13.4	67.7 ± 11.7
M	45.9 ± 25.3	39.4 ± 11.8	31.2 ± 12.7	31.0 ± 11.0
Lg	0.7 ± 0.6	0.8 ± 0.6	1.5 ± 0.9	1.3 ± 0.7

**Table 6 ijms-24-16647-t006:** Relative frequency of different sizes of PNX^+^/nNOS^+^ perikarya found in DRGs of the C, Th, L and S spinal cord segments. The obtained data were pooled for each animal and are presented as mean ± SD (N = 6 animals).

	Segments of the Spinal Cord (%)			
Size of DRG Cells	C	Th	L	S
Sl	38.9 ± 15.2	43.4 ± 9.3	51.0 ± 3.4	50.9 ± 4.8
M	58.6 ± 15.1	56.1 ± 9.0	47.1 ± 4.6	47.9 ± 5.0
Lg	2.5 ± 3.3	0.5 ± 0.3	1.9 ± 1.2	1.2 ± 1.1

**Table 7 ijms-24-16647-t007:** The relative frequency of different sizes of PNX^+^/GAL^+^ perikarya found in DRGs of the C, Th, L and S spinal cord segments. The obtained data were pooled for each animal and are presented as mean ± SD (N = 6 animals).

	Segments of the Spinal Cord (%)			
Size of DRG Cells	C	Th	L	S
Sl	99.3 ± 1.0	97.3 ± 3.3	99.7 ± 0.5	100.0
M	0.7 ± 1.0	2.7 ± 3.3	0.3 ± 0.5	0.0
Lg	0.0	0.0	0.0	0.0

**Table 8 ijms-24-16647-t008:** The relative frequency of different sizes of PNX^+^/CRT^+^ perikarya found in DRGs of the C, Th, L and S spinal cord segments. The obtained data were pooled for each animal and are presented as mean ± SD (N = 6 animals).

	Segments of the Spinal Cord (%)			
Size of DRG Cells	C	Th	L	S
Sl	80.3 ± 2.6	98.1 ± 3.3	98.9 ± 1.9	99.5 ± 0.8
M	17.2 ± 2.5	1.8 ± 3.1	1.0 ± 1.7	0.5 ± 0.8
Lg	2.5 ± 4.1	0.1 ± 0.2	0.1 ± 0.1	0.0

**Table 9 ijms-24-16647-t009:** Relative frequency of different sizes of PNX^+^/PACAP^+^ perikarya found in DRGs of the C, Th, L and S spinal cord segments. The obtained data were pooled for each animal and are presented as mean ± SD (N = 6 animals); the data were statistically analyzed using the Student’s two-tailed *t*-test for unpaired data and using one-way ANOVA with mean values compared using Tukey’s multiple comparison test. Asterisks mark statistically significant differences: * *p* < 0.05, ** *p* = 0.005. The results were statistically analyzed using GraphPad PRISM 8.0 software.

	Segments of the Spinal Cord (%)			
Size of DRG Cells	C	Th	L	S
Sl	80.7 ± 7.7 *	94.7 ± 2.1 *; **	90.3 ± 5.6	83.3 ± 2.7 **
M	16.6 ± 5.6 *	5.3 ± 2.1 *; **	9.7 ± 5.6	16.7 ± 2.7 **
Lg	2.7 ± 2.3	0.0	0.0	0.0

**Table 10 ijms-24-16647-t010:** The relative frequency of different sizes of PNX^+^/CART^+^ perikarya found in DRGs of the C, Th, L and S spinal cord segments. The obtained data were pooled for each animal and are presented as mean ± SD (N = 6 animals).

	Segments of the Spinal Cord (%)			
Size of DRG Cells	C	Th	L	S
Sl	20.4 ± 6.2	66.6 ± 2.8	41.1 ± 5.5	49.2 ± 2.7
M	79.6 ± 6.2	33.4 ± 2.8	56.4 ± 1.0	50.8 ± 2.7
Lg	0.0	0.0	2.5 ± 1.4	0.0

**Table 11 ijms-24-16647-t011:** The relative frequency of different sizes of PNX^+^/SOM^+^ perikarya found in DRGs of the C, Th, L and S spinal cord segments. The obtained data were pooled for each animal and are presented as mean ± SD (N = 6 animals).

	Segments of the Spinal Cord (%)			
Size of DRG Cells	C	Th	L	S
Sl	82.5 ± 2.1	81.9 ± 2.3	86.9 ± 4.5	89.6 ± 9.0
M	17.5 ± 2.1	17.7 ± 1.7	12.8 ± 2.0	10.4 ± 9.0
Lg	0.0	0.4 ± 0.5	0.3 ± 1.8	0.0

**Table 12 ijms-24-16647-t012:** List of primary antisera and secondary reagents used in the study: CART, CGRP, CRT, GAL, nNOS, PACAP, protein gene product 9.5 (PGP 9.5), PNX, SOM, SP, FITC, CY3.

Antigen	Code	Dilution	Host	Supplier
Primary antibodies				
CART	MAB 163	1:1000	Mouse	R&D Systems, Wiesbaden, Germany
CGRP	T-5027	1:800	Guinea pig	Peninsula Laboratories, San Carlos, CA, USA
CRT	6B3	1:2000	Mouse	SWANT, Burgdorf, Switzerland
GAL	T-5036	1:1500	Guinea pig	Peninsula Laboratories, San Carlos, CA, USA
nNOS	N2280	1:200	Mouse	Sigma-Aldrich, St. Louis, MO, USA
PACAP	T-5039	1:1000	Guinea pig	Peninsula Laboratories, San Carlos, CA, USA
PGP 9.5	7863-2004	1:5000	Mouse	Biogenesis, Poole, UK,
PNX	H-079-01	1:7000	Rabbit	Phoenix Pharmaceuticals Inc., Burlingame, CA, USA,
SOM	MAB 354	1:50	Rat	Merck Millipore, Temecula, CA, USA
SP	8450-0004	1:200	Rat	Bio-Rad, Kidlington, UK
Secondary reagents				
CY3-conjugated anti-rabbit	711-166-152	1:700	Donkey	Jackson I.R.; West Grove, PA, USA, Baltimore Pike
FITC-conjugated anti-mouse IgG	715-096-151	1:800	Donkey	Jackson I.R.; West Grove, PA, USA, Baltimore Pike
FITC-conjugated anti-guinea pig IgG	706-095-148	1:800	Donkey	Jackson I.R.; West Grove, PA, USA, Baltimore Pike
FITC-conjugated anti-rat IgG	712-095-153	1:400	Donkey	Jackson I.R.; West Grove, PA, USA, Baltimore Pike

**Table 13 ijms-24-16647-t013:** Double staining combinations employed in this study.

Primary Antibody	Primary Antibody	Secondary Antibody
PNX (rabbit)	CART (mouse)	CY3 (rabbit), FITC (mouse)
PNX (rabbit)	CGRP (guinea pig)	CY3 (rabbit), FITC (guinea pig)
PNX (rabbit)	CRT (mouse)	CY3 (rabbit), FITC (mouse)
PNX (rabbit)	GAL (guinea pig)	CY3 (rabbit), FITC (guinea pig)
PNX (rabbit)	nNOS (mouse)	CY3 (rabbit), FITC (mouse)
PNX (rabbit)	PACAP (guinea pig)	CY3 (rabbit), FITC (guinea pig)
PNX (rabbit)	PGP (mouse)	CY3 (rabbit), FITC (mouse)
PNX (rabbit)	SOM (rat)	CY3 (rabbit), FITC (rat)
PNX (rabbit)	SP (rat)	CY3 (rabbit), FITC (rat)

**Table 14 ijms-24-16647-t014:** List of antigens used in pre-absorption test.

Antigen	Code	Dilution	Supplier
CART	MAB0041	1:1000	R&D Systems, Germany
CGRP	T-4030	1:800	Peninsula Laboratories, San Carlos, CA, USA
CRT	6-His human calretinin (recombinant) Lot No.: 22	1:2000	SWANT, Switzerland
GAL	T-4862	1:1500	Peninsula Laboratories, San Carlos, CA, USA
nNOS	N3033	1:200	Sigma, St. Louis, MO, USA
PACAP	A9808	1:1000	Sigma, St. Louis, MO, USA
PNX	079-01	1:7000	Phoenix Pharmaceuticals Inc.; Burlingame; Kalifornia; San Carlos, CA, USA
SOM	S9129	1:50	Sigma-Aldrich, St. Louis, MO, USA
SP	S6883	1:200	Sigma-Aldrich, St. Louis, MO, USA

## Data Availability

The data that support the findings of this study are available from the corresponding author upon reasonable request.

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
