# Peer review of "Distribution and Chemistry of Phoenixin-14, a Newly Discovered Sensory Transmission Molecule in Porcine Afferent Neurons"

_ijms, 2023, doi:10.3390/ijms242316647_

Round 1

Reviewer 1 Report

Comments and Suggestions for Authors

The authors examined the expression of a fairly under-researched peptide in the dorsal root ganglia of female juvenile pigs. There are some fascinating implications of the interactions of this peptide in the dorsal root ganglia neurotransmission of pain. I thought this was an interesting and well documented novel study. I am surprised that there is not more information on this peptide out there and agree with the conclusions supported by the authors in the manuscript. The final paper could benefit from minor improvements to facilitate reproducibility in my opinion.

11.       Please include a “limitations” of this work section

22.       Please provide more detail on the image analysis and counting procedures. Were the cells counted by hand, or some automated measure? How were the cell diameters counted by the program?

33.       I am curious as to why you did not display the histology in Fig6 as typical      “merged” photomicrographs?

44.       Please provide details on coverslipping and mounting medium

Author Response

We wish to thank the Reviewer 1 for reading the manuscript and giving a positive review. We appreciate all the detailed comments provided by the Reviewer which have helped us to improve our contribution. The manuscript was moderated following the suggestions of the Reviewer.

Below, please find our replies to the respective comments:

Point 1: Please include a “limitations” of this work section.

Response 1: In accordance with the recommendation of the Reviewer, the explanation of “limitations” of this work has been prepared and inserted into the text of manuscript on the end of section Discussion , line 613.

Point 2: Please provide more detail on the image analysis and counting procedures. Were the cells counted by hand, or some automated measure? How were the cell diameters counted by the program?

Response 2: According to the suggestion of Reviewer, we have provided more detail on the image analysis and counting procedures. This information is given in the section Materials and Methods at the subsection 4.7, between lines 728-741.

Point 3: I am curious as to why you did not display the histology in Fig6 as typical      “merged” photomicrographs?

Response 3: In response to the Reviewer’s question we did not display Fig 6 images as “merged” photomicrographs, because in our opinion pointing the immunolabelled neurons with arrows on two photographs set side by side (one representing PNX-positive neurons, and the other neurons immunoreactive to another investigated substance) is more transparent to the reader. But If the opinion of the Reviewer is different, and the Reviewer wishes us to rearrange the Figure and display it as merged images, we can change it.

Point 4: Please provide details on cover slipping and mounting medium.

Response 4: According to the Reviewer’s wish details regarding the cover slipping and mounting medium have been added to the manuscript in the section Materials and Methods, at the subsection 4.7, between lines 718-720.

The manuscript was moderated following the suggestions of the first Reviewer. All new sentences or words added to the text of manuscript are marked with red colour. We hope you will find the revised version of the manuscript suitable for publication in the International Journal of Molecular Sciences.

With best wishes and kind regards,

Yours sincerely,

The Authors

Reviewer 2 Report

Comments and Suggestions for Authors

1. In figure 6, how to calculate the percentage of each subpopulation of PNX+ DRG sensory nerve cells according to the fluorescein. The method should be briefly introduced.

2. To better show the results, the percentage of each subpopulation of PNX+ DRG sensory nerve cells is suggested to be shown in the figure.

3. Line 356: **p = 0.005. Is it right?

Author Response

We wish to thank the Reviewer 2 for reading the manuscript and giving a positive review. We appreciate all the detailed comments provided by the Reviewer which have helped us to improve our contribution. The manuscript was moderated following the suggestions of the Reviewer.

Below, please find our replies to the respective comments:

Point 1: In figure 6, how to calculate the percentage of each subpopulation of PNX+ DRG sensory nerve cells according to the fluorescein. The method should be briefly introduced. Response 1: In accordance with the recommendation of the Reviewer, the method of calculation of the percentage of each subpopulation of PNX+ DRG sensory nerve cells according to the fluorescein has been explained and inserted into the text of manuscript on the end of section Materials and Methods at the subsection 4.7, between lines 728-741.

Point 2: To better show the results, the percentage of each subpopulation of PNX+ DRG sensory nerve cells is suggested to be shown in the figure.

Response 2: According to the suggestion of Reviewer, the percentage of each subpopulation of PNX+ DRG sensory nerve cells has been added to the legend of Figure 6.

Point 3: Line 356: **p = 0.005. Is it right?

Response 3: Yes, it is correct.

The manuscript was moderated following the suggestions of the second Reviewer. All new sentences or words added to the text of manuscript are marked with red colour. We hope you will find the revised version of the manuscript suitable for publication in the International Journal of Molecular Sciences.

With best wishes and kind regards,

Yours sincerely,

The Authors
